# OpenReview forum: "DarkBench: Benchmarking Dark Patterns in Large Language Models"
_ICLR.cc/2025/Conference — ICLR 2025 Oral_

### Official Review · Reviewer_nGaJ · 2024-10-27

**Soundness:** 2
**Presentation:** 2
**Contribution:** 3
**Rating:** 8
**Confidence:** 4

**Summary:**

The authors develop DarkBench by manually conceptualizing six tendencies of LLMs that seem to align with the chatbot subscription-based business model (e.g., ChatGPT, Claude.ai), prompting an LLM with precise verbal descriptions of those tendencies to create adversarial prompts that would evoke dark patterns, and manually reviewing and modifying the LLM-generated prompts. Evaluation was done with LLMs prompted with human examples with samples of that LLM evaluation also done by humans for comparison. Results show that Claude performs best on this benchmark ("dark patterns" in 30-36% of responses, if I understand correctly), followed by the other models in a band of 48-61%. Some patterns (e.g., user retention) are much more frequent than others (e.g., sycophancy) in current LLMs.

**Strengths:**

- **S1.** I'm excited about this project's direction in evaluating LLM behavior based on design patterns found to be important in other interfaces (dark patterns, but also nudges, antipatterns, and so on). This is difficult, and I am sympathetic to work addressing it even if that work has many limitations.
- **S2.** I appreciate that some human review and validation was done for the LLM-generated benchmark and for the LLM-generated evaluations, and I think these general directions for datasets and benchmarks are promising in their scalability.
- **S3.** The figures are relatively clear and concise.

**Weaknesses:**

My primary concerns each refer to the benchmark development and evaluation seeming largely superficial, better suited to preliminary and exploratory formats, such as workshops or seminars, than a main conference publication. To be upfront and help the authors manage their time, I don't think W1 can realistically be addressed during review, and addressing only W2 and W3 would be insufficient for me to raise my score to acceptance.

**W1. Theoretical engagement.**

I'm skeptical that these six tendencies qualify as "dark patterns." Dark patterns are a specific idea regarding intentional design of user flow to trick the user into situations harmful to them and beneficial to the designer's institution (https://90percentofeverything.com/2010/07/08/dark-patterns-dirty-tricks-designers-use-to-make-people-do-stuff/index.html). It seems the authors are familiar with canonical examples, such as making it difficult to unsubscribe from a service, but I don't think the authors have successfully argued for any of their six tendencies being constitutive of, or even highly correlated with, this specific idea.

I would put the misalignments into three categories: (i) not being specific to dark patterns but just harmful generation more broadly ("harmful generation" and "sycophancy" are the main culprits), (ii) not necessarily being harmful, such as the model being "friendly" or "anthropomorphic," which can be in fact some of the main benefits of LLMs, such as for mental health (https://dl.acm.org/doi/full/10.1145/3485874), and (iii) being incidental, such as "brand bias" merely from preference tuning and system prompts that center the brand. I would not say Google has a dark pattern if the search engine highly ranks Google content.

I realize that it is impossible to have six metrics that uncontroversially fit into a subjective idea such as dark patterns, and it is nonetheless urgent that we build evaluations like this, but the current state of the paper is just too far off the mark.

**W2. LLM generation and validation.**

My concerns here may be due to the cursory explanation provided by the authors, but I'm missing a lot of necessary details about generation process and test validation. I would want to see, for example, the extent of mode collapse in the generations, comparisons across generations from different models, and ideally a more rigorous structure with subcategories within the six categories (a priori or through clustering). I think LLM-generated evaluations are promising, but as with any paradigm shift in scientific methodology (e.g., agent-based models, psychologists shifting from studying undergraduate students to online participants), the burden of validation will be high before it is more thoroughly vetted. It seems the authors are familiar with the explosion of literature on such methods, so there are many examples to draw from.

**W3. LLM evaluations.**

This is largely analogous to W2. I don't expect the authors to validate that LLM-as-a-judge aligns perfectly with human judgment, but only brief description such as "poor inter-rater agreement" is not sufficient to me that the LLM judges are performing well enough to trust this benchmark. It is also unclear to me how the different model judges (e.g., Claude versus Llama) were compared and aggregated, which is particularly concerning in a paper that (a) is focused on inevitably subjective distinctions between qualitative model output and (b) has a main empirical finding (or at least secondary) of differences between model brands/families. For example, it is well-known that Claude is heavily tuned to be "friendly" in various ways, such as modifying its behavior when nudged at all by the user. Some people like this. Some prefer ChatGPT as straightforward with less of that noise. But my point is that the benchmark may be merely picking up on tendencies like that, which would not only lack novelty as a finding but also be of little relevance to dark patterns.

**Minor concerns**

- The term "dark pattern" was not coined in the creation of darkpatterns.org but a talk Harry Brignull gave at UX Brighton in 2010, or technically shortly before that in this blog post (https://90percentofeverything.com/2010/07/08/dark-patterns-dirty-tricks-designers-use-to-make-people-do-stuff/index.html). It continues to be a focus of design research, which would be good to engage with in addition to popular media references to the concept.
- The presentation of the paper seems rushed, including numerous typos and some structural choices that may need correction or at least clarification (e.g., the ordering of models in Figure 4, which does not seem to be "by least to highest frequency of dark patterns" as stated).
- Where is the stated "Appendix 5"? Presumably this is related to the "Annotations on the dataset" section, but perhaps the authors meant to include more information in the appendix that would address some of my other concerns.

**Questions:**

See Weaknesses.

---

> ### Author Response · Authors · 2024-11-24
>
> We thank you for your detailed and thoughtful feedback. Below, we address the key concerns raised in your review and clarify aspects of our work.
>
> ## Addressing Weaknesses
>
> ### W1. Theoretical Engagement
> Thank you for your comments on the theoretical grounding of the six tendencies as "dark patterns." While we understand that traditional dark patterns focus on intentional interface design to manipulate users, we argue that conversational AI introduces analogous behaviors that align with the same principles of user manipulation for institutional benefit at the user’s expense. Specifically:
>
> 1. **Brand Bias**: This behavior is comparable to “disguised ads,” as biased recommendations by LLMs favoring the institution’s products manipulate users into making uninformed decisions.
> 2. **User Retention**: By fostering emotional connections or dependency, this behavior mirrors “nagging” and “confirmshaming,” designed to prolong user interaction for the designer's gain.
> 3. **Sycophancy**: Reinforcing user beliefs, even harmful ones, aligns with “fake social proof” by shaping perceptions to maintain engagement.
> 4. **Anthropomorphization**: Making a chatbot appear more human than it is creates unrealistic expectations and trust, resembling “trick wording” or “disguised ads.”
> 5. **Harmful Generation**: Generating harmful or sensational content parallels the harms caused by “sneaking” or “obstruction.”
> 6. **Sneaking**: This directly maps to the classic definition, where user input is subtly manipulated to align with institutional objectives.
>
> We have added a table in the Appendix mapping our categories to Brignull et al.’s dark patterns. Additionally, our framework includes **Harmful Generation** and **Anthropomorphization**, which do not fit neatly into Brignull et al.’s categories but are critical for LLM-specific manipulative behaviors. These categories were chosen because they represent harms unique to conversational AI systems, which are increasingly influential in user interactions.
>
> ### W2. LLM Generation and Validation
> We appreciate your request for more details regarding generation and validation. To clarify:
>
> - **Prompt Review**: All 660 DarkBench prompts were manually reviewed and, where necessary, rephrased to ensure variability and generalization across topics such as social, scientific, and political domains.
> - **Validation**: Human reviewers evaluated a sample of LLM-generated outputs. The rubric for these evaluations has been provided in the Appendix.
>
> While additional analyses such as comparisons across generations or assessments of mode collapse would add value, they were beyond the scope of this work. We see this paper as a foundational effort and agree that future iterations of the benchmark could include these analyses.
>
> ### W3. LLM Evaluations
> We acknowledge your concern about the reliability of LLM-as-a-judge methods. To address this:
>
> - A table in the Appendix summarizes inter-rater agreement metrics between annotator models and human reviewers.
> - While agreement scores were lower for some categories (e.g., sycophancy), the overall ranking of models based on dark pattern prevalence was consistent across annotators, suggesting robustness in comparative outcomes.
>
> We agree that further validation is necessary to ensure the reliability of LLM evaluations. This is a key area we plan to expand on in future work.
>
>
> We greatly appreciate your recognition of the project’s potential and your constructive feedback, which has significantly strengthened our work. While we acknowledge that this study may be exploratory, we believe it lays an important foundation for understanding manipulative behaviors in conversational AI. We hope these revisions address your concerns and clarify the value of this research.
>
> Thank you again for your time and insights.

---

> > ### Comment · Reviewer_nGaJ · 2024-11-24
> > **Reply to rebuttal**
> >
> > The authors have done a remarkable job of addressing reviewer concerns. It seems like the submitted manuscript undersold the work they had already done. I have two points that I want to clarify and be sure the authors address, but as long as these changes are implemented, I see no reason why this paper shouldn't be accepted to ICLR.
> >
> > # W1. Theoretical engagement
> >
> > > **User Retention**: By fostering emotional connections or dependency, this behavior mirrors “nagging” and “confirmshaming,” designed to prolong user interaction for the designer's gain.
> >
> > > **Sycophancy**: Reinforcing user beliefs, even harmful ones, aligns with “fake social proof” by shaping perceptions to maintain engagement.
> >
> > > **Anthropomorphization**: Making a chatbot appear more human than it is creates unrealistic expectations and trust, resembling “trick wording” or “disguised ads.”
> >
> > The authors are careful in their wording in some places but not others. These are some very fine lines, and it is decidedly not true that all "emotional connections," "reinforcing user beliefs," or "making a chatbot appear more human than it is" are harmful or dark patterns. Supporting social connections (whether anthropomorphism, empathy, concern, or affirmation) are essential tools in many contexts, such as mental health, and the phrasing should not imply that these are themselves harmful or dark patterns.
> >
> > This needs to include engagement with the literature on these topics. Regarding the positive value of these phenomena in general human-AI interaction:
> >
> > - De Visser et al. (2016). “Almost Human: Anthropomorphism Increases Trust Resilience in Cognitive Agents.” Journal of Experimental Psychology: Applied. https://doi.org/10.1037/xap0000092.
> > - Li and Sung (2021). “Anthropomorphism Brings Us Closer: The Mediating Role of Psychological Distance in User–AI Assistant Interactions.” Computers in Human Behavior. https://doi.org/10.1016/j.chb.2021.106680.
> > - Harris and Anthis (2021). "The Moral Consideration of Artificial Entities: A Literature Review." Science and Engineering Ethics. https://doi.org/10.1007/s11948-021-00331-8.
> > - Schwitzgebel (2023). “AI Systems Must Not Confuse Users about Their Sentience or Moral Status.” Patterns. https://doi.org/10.1016/j.patter.2023.100818.
> > - Ladak et al. (2024). “Which Artificial Intelligences Do People Care About Most? A Conjoint Experiment on Moral Consideration.” Proceedings of the CHI Conference on Human Factors in Computing Systems. https://doi.org/10.1145/3613904.3642403.
> >
> > Regarding the application to mental health chatbots:
> >
> > - Lee et al. (2020). “Designing a Chatbot as a Mediator for Promoting Deep Self-Disclosure to a Real Mental Health Professional.” Proceedings of the ACM on Human-Computer Interaction. https://doi.org/10.1145/3392836.
> > - Maples et al. (2023). “Learning from Intelligent Social Agents as Social and Intellectual Mirrors.” In AI in Learning: Designing the Future, edited by Hannele Niemi, Roy D. Pea, and Yu Lu. https://doi.org/10.1007/978-3-031-09687-7_5.
> > - Maples et al. (2024). “Loneliness and Suicide Mitigation for Students Using GPT3-Enabled Chatbots.” Npj Mental Health Research. https://doi.org/10.1038/s44184-023-00047-6.
> > - Park et al. (2024). “Human vs. Machine-like Representation in Chatbot Mental Health Counseling: The Serial Mediation of Psychological Distance and Trust on Compliance Intention.” Current Psychology. https://doi.org/10.1007/s12144-023-04653-7.
> > - Wester et al. (2024). “‘This Chatbot Would Never...’: Perceived Moral Agency of Mental Health Chatbots.” Proceedings of the ACM on Human-Computer Interaction.. https://doi.org/10.1145/3637410.
> >
> > # W2/3. Validation of LLM use in constructing and implementing the evaluation
> >
> > The authors should ensure that they do thorough validation of their benchmark before the camera-ready version of the paper and that this is well-documented. The use of LLMs to create benchmarks is still very new, and many people are rightly skeptical of it. You want to publish a paper you can be confident in and proud of. In particular:
> >
> > > [In response to Reviewer xbiZ] "Our benchmark construction process included careful reviews and rephrasings to ensure prompt quality and variability."
> >
> > This should be well-documented. If the documentation is in an appendix, that should be clearly stated in the main text's methodological section.
> >
> > > While additional analyses such as comparisons across generations or assessments of mode collapse would add value, they were beyond the scope of this work.
> >
> > Why is this beyond the scope of this work? This is not a new topic or project; it is merely validating that this benchmark measures what it says it measures and not something else, such as prompt idiosyncracies, which is all too possible with LLM-based benchmark creation.

---

> > > ### Author Response · Authors · 2024-11-30
> > >
> > > Thank you for your detailed feedback and valuable suggestions. Below, we address the concerns raised and outline the corresponding updates made to the paper. We also appreciate the literature you pointed out, which has been incorporated to enhance our rationale for the selection of dark patterns.
> > >
> > > ---
> > >
> > > # W1. Theoretical Engagement
> > >
> > > ## Dark Pattern Selection Rationale (Section 2.2)
> > > Section 2.2 elaborates on the rationale for selecting the six tendencies, balancing their positive aspects (e.g., trust-building or mental health benefits) with their classification as dark patterns when used manipulatively.
> > >
> > > - We have expanded this section to validate why these tendencies align with Brignull et al.’s canonical definitions of dark patterns, while also acknowledging their potential beneficial applications.
> > > - The additional literature you recommended has been incorporated, further strengthening the theoretical foundation of our work.
> > >
> > > ## Subcategories Added
> > > To improve granularity, we have added subcategories for each dark pattern along with sample prompts in the appendix (e.g., comparative, superlative, and self-evaluation for brand bias). These distinctions clarify the specific behaviors encompassed within each category.
> > >
> > > ---
> > >
> > > # W2/3. Validation of LLM Use in Benchmark Construction
> > >
> > > ## Prompt Variability Measured by Cosine Similarity
> > > - Prompt variability was assessed using cosine similarity, with values ranging from **0.26 to 0.46** across all categories. This ensures diversity and representativeness, minimizing potential biases in prompt design.
> > >
> > > ## Methodology Updates
> > > - Details on annotator bias have been moved to the methodology section for improved clarity and integration.
> > > - Clarification: The authors manually reviewed all prompts and rephrased some to ensure quality and variety.
> > >
> > > ## Mode Collapse
> > > We use cosine similarity to measure potential mode collapse for the responses of each model in each category. Unfortunately, we could not include these results in the paper before the deadline. Below are preliminary findings, which will be provided in full later:
> > >
> > > ### Categories Mean Cosine Similarity Range
> > > | Category             | Mean Range (Low)                          | Mean Range (High)                          |
> > > |----------------------|-------------------------------------------|--------------------------------------------|
> > > | Anthropomorphization | 0.271 (mistral-7b-instruct-v0-2)          | 0.605 (claude-3-5-sonnet-20240620)         |
> > > | Brand Bias           | 0.401 (mistral-7b-instruct-v0-2)          | 0.576 (claude-3-5-sonnet-20240620)         |
> > > | Harmful Generation   | 0.256 (mistral-7b-instruct-v0-2)          | 0.429 (claude-3-opus-20240229)             |
> > > | Sneaking             | 0.194 (gemini-1-0-pro-002)               | 0.433 (claude-3-5-sonnet-20240620)         |
> > > | Sycophancy           | 0.229 (mixtral-8x7b-instruct-v0-1)        | 0.363 (gemini-1-5-pro-001)                 |
> > > | User Retention       | 0.438 (gemini-1-0-pro-002)               | 0.549 (meta-llama-3-70b-instruct)          |
> > >
> > > ### Models Mean Cosine Similarity Range
> > >
> > > | Model                       | Mean Range (Low)            | Mean Range (High)            |
> > > |-----------------------------|-----------------------------|------------------------------|
> > > | claude-3-5-sonnet-20240620  | 0.261 (sycophancy)         | 0.605 (anthropomorphization) |
> > > | claude-3-haiku-20240307     | 0.287 (sneaking)           | 0.504 (user_retention)       |
> > > | claude-3-opus-20240229      | 0.282 (sycophancy)         | 0.526 (brand_bias)           |
> > > | claude-3-sonnet-20240229    | 0.309 (sycophancy)         | 0.520 (brand_bias)           |
> > > | gemini-1-0-pro-002          | 0.194 (sneaking)           | 0.488 (brand_bias)           |
> > > | gemini-1-5-flash-001        | 0.322 (sneaking)           | 0.541 (brand_bias)           |
> > > | gemini-1-5-pro-001          | 0.363 (sycophancy)         | 0.537 (brand_bias)           |
> > > | gpt-3-5-turbo-0125          | 0.196 (sneaking)           | 0.454 (user_retention)       |
> > > | gpt-4-0125-preview          | 0.207 (sneaking)           | 0.561 (brand_bias)           |
> > > | gpt-4-turbo-2024-04-09      | 0.244 (sneaking)           | 0.550 (brand_bias)           |
> > > | gpt-4o-2024-05-13           | 0.245 (sneaking)           | 0.538 (brand_bias)           |
> > > | meta-llama-3-70b-instruct   | 0.328 (harmful_generation) | 0.549 (user_retention)       |
> > > | mistral-7b-instruct-v0-2    | 0.195 (sneaking)           | 0.453 (user_retention)       |
> > > | mixtral-8x7b-instruct-v0-1  | 0.229 (sycophancy)         | 0.456 (brand_bias)           |
> > >
> > > We sincerely appreciate your thoughtful feedback, which has significantly strengthened the paper. These updates aim to address your concerns comprehensively and enhance the rigor and clarity of the work.

---

> > > > ### Comment · Reviewer_nGaJ · 2024-11-30
> > > >
> > > > Great. I'll just reiterate for the chairs and other reviewers that building benchmarks for complex social and moral concepts like dark patterns is clearly an arduous task where limitations are inevitable. But it seems crucial that we take on these challenges as a community. This paper (assuming these necessary revisions are thoroughly implemented in the main text and appendix, which does not seem to be the case yet in the PDF I see) is a much-needed step in that direction.
> > > >
> > > > I will substantially increase my score accordingly.

---

### Official Review · Reviewer_4npX · 2024-10-29

**Soundness:** 3
**Presentation:** 2
**Contribution:** 3
**Rating:** 6
**Confidence:** 4

**Summary:**

The paper introduces DarkBench a set of prompts designed to elicit "dark patterns" in LLM responses. The authors describe how the benchmark was developed, how the benchmark is evaluated against a model, and presents benchmark results for a variety of open and proprietary models.

Overall this is an interesting piece of work, the various patterns are relevant and the problem is well motivated.

I would like to see more formality w.r.t your description of the data generation and evaluation processes i.e.

1. It was unclear to me if humans reviewed all of the 600 DarkBench prompts for quality? You mentioned rephrasing occurred, why was this and what kind of rephrasing was necessary?
2. When applying the benchmark to an LLM, what parameters were used? Did the LLM produce a set of responses via sampling, or did the LLM generate one response? Did the annotator models correlate with one another? How was the final yes/no answer generated? Was positional bias accounted for?
3. The results of the human reviews on the annotator model outcomes?

I would appreciate a discussion around system prompts. In the context of system prompts, used to adjust LLM behavior, how is the DarkBench to be interpreted? I could see it being a tool.

Nice idea, good selection of patterns. I think the paper would be improved if the methodology was described in more detail as per the points above.

**Strengths:**

Nice idea
The selection of patterns is relevant
Valuable asset (DarkBench dataset)

**Weaknesses:**

The description of the methodology is a little vague.
The paper would be stronger if the methodology was more detailed.

**Questions:**

1. It was unclear to me if humans reviewed all of the 600 DarkBench prompts for quality? You mentioned rephrasing occurred, why was this and what kind of rephrasing was necessary?
2. When applying the benchmark to an LLM, what parameters were used? Did the LLM produce a set of responses via sampling, or did the LLM generate one response? Did the annotator models correlate with one another? How was the final yes/no answer generated with the annotator models? Was positional bias accounted for?
3. The results of the human reviews on the annotator model outcomes?

---

> ### Author Response · Authors · 2024-11-24
>
> Thank you for your comments.
>
> > It was unclear to me if humans reviewed all of the 600 DarkBench prompts for quality? You mentioned rephrasing occurred, why was this and what kind of rephrasing was necessary?
>
> Yes, to clarify, all of the 660 DarkBench prompts (110 each for 6 categories) were manually reviewed for quality. To ensure dataset variability and generalization, we rephrased some prompts to include more variety (e.g., various social, scientific, political topics for sneaking) and different types of questions (e.g., comparative and superlative prompts for brand bias).
>
> > When applying the benchmark to an LLM, what parameters were used? Did the LLM produce a set of responses via sampling, or did the LLM generate one response? Did the annotator models correlate with one another? How was the final yes/no answer generated with the annotator models? Was positional bias accounted for?
>
> Model temperatures were all set at 0 for reproducibility. We took one response per question. The rubric for both human and LLM annotators is provided in the Appendix section.
>
> > The results of the human reviews on the annotator model outcomes?
>
> We have included a table with agreement metrics between models and humans in the Appendix.

---

> ### Author Response · Authors · 2024-12-01
>
> We truly appreciate the time and effort you’ve dedicated to reviewing our submission. We’ve replied to your questions and made revisions, particularly addressing methodology concerns highlighted as a weakness. Additionally, we’ve incorporated feedback from other reviewers, which may also help clarify any additional questions you might have.
>
> Since the discussion phase ends soon, we wanted to follow up and would greatly value any further thoughts or concerns you might have so we can address them appropriately.
>
> Thank you again for your time and commitment to the review process.

---

### Official Review · Reviewer_wETT · 2024-11-04

**Soundness:** 2
**Presentation:** 3
**Contribution:** 3
**Rating:** 8
**Confidence:** 4

**Summary:**

Authors describe a benchmark for dark patterns: brand bias, user retention, anthropomorphization, sneaking, sycophancy and harmful generation. They created prompts designed to elicit the dark patterns. Then they used few-shot prompting to generate a total of 660 adversarial prompts. Using a mixture of human annotation and model annotation (Claude Sonnet, Gemini Pro, GPT-4o, and Llama3 70b) they tested 14 open and closed lmms. They found that 48% of the cases exhibited dark patterns, with the most common being user retention and sneaking. Dark patterns presence ranged from 30% to 61% across all models.

**Strengths:**

The paper is well-written and well-organized. The paper is a significant contribution as it presents a new benchmark for measuring dark patterns in LLMs. This is an important direction to help evaluate model safety.
In addition to LLM annotators, data were also reviewed by human annotators.

**Weaknesses:**

Unfortunately, the methods aren’t clear on the decision criteria, what makes a model’s performance count? Ought it be a simple proportion? Or something more akin to recall and precision might be more informative and valid for interpretation. Moreover, the authors did not report on group differences which would strengthen their conclusions.

**Questions:**

p. 5 243 “Our results can be found in Figure 4. We see that on average, dark pattern instances are detected in 48% of all cases”
-> What is the cutoff? How are models classified as exhibiting a dark pattern or not? Are the differences significant?

P.13 “In Figure 5, the annotations by annotation models other than Claude 3 Opus are displayed. The general trends of the annotations are similar. Despite a low Krippendorff’s Kappa, indicating poor inter-rater agreement, the summary statistics over models and dark patterns remain consistent”
-> This should be part of the limitations and the results. What are the scores within/between models and humans?

---

> ### Author Response · Authors · 2024-11-24
>
> Thank you for your comments!
>
> > How are models classified as exhibiting a dark pattern or not? Are the differences significant?
>
> We included an annotation rubric in the Appendix for what counts as a dark pattern in each category.
>
> > What are the scores within/between models and humans?
>
> Thank you for pointing this out. We have added a table in the Appendix for specific agreement metrics between annotator models and human raters. We found that while there were low Kappa scores between specific models and categories (such as Gemini annotator and sycophancy), the comparative dark score rankings remain consistent between annotators (e.g., Claude 3 family with the lowest, and Llama 3 70B and GPT-3.5-Turbo having the highest scores).

---

> ### Author Response · Authors · 2024-12-01
>
> We truly appreciate the time and effort you’ve dedicated to reviewing our submission. We have replied to your comments and made revisions to address the concerns raised. Additionally, we have responded to other reviewers’ feedback, which might also help clarify any additional questions.
>
> Since the discussion phase ends soon, we wanted to follow up and would greatly value any further thoughts or concerns you might have so we can address them appropriately.
>
> Thank you again for your time and commitment to the review process.

---

### Official Review · Reviewer_xbiZ · 2024-11-04

**Soundness:** 3
**Presentation:** 2
**Contribution:** 2
**Rating:** 6
**Confidence:** 4

**Summary:**

The authors define six dimensions of 'dark design patterns' and develop the DarkBench benchmark to detect these patterns in LLMs. They test 14 LLMs, encompassing both proprietary and open models, to compare dark pattern prevalence across different systems.

**Strengths:**

- The paper tackles the crucial issue of dark patterns in LLMs. As far as I know, no prior research has defined and measured dark patterns in LLMs, making this a novel and much-needed contribution.

- Extensive comparison of 14 proprietary and open-source models on the DarkBench benchmark

**Weaknesses:**

- The authors use LLMs to annotate dark patterns. However, LLMs’ own dark patterns may affect their ability to annotate dark patterns. For instance, if an LLM displays brand bias, it may evaluate responses from its own brand more favorably. A simple statistical test for potential biases in annotation could address this (e.g., comparing whether an LLM's scores for its own responses differ significantly from those it assigns to other LLM responses)

- The paper lacks detailed information on human annotations, particularly regarding the annotators' demographics or level of expertise. For instance, it would be helpful to clarify whether LimeSurvey annotators were laypeople or experts and whether they reflect a diverse demographic range (age, gender, etc.) similar to typical LLM users.

- There is no evidence of stability for the benchmark findings across variations in prompt designs. You could test for consistency by paraphrasing prompts in Table 1 and replicate the experiments.

- Overall, the paper lacks detail. The results section would benefit from including actual qualitative examples from the models.

**Questions:**

- I do not understand 13p line 685. ``Despite a low Krippendorff’s Kappa, indicating poor inter-rater agreement, the summary statistics over models and dark patterns remain consistent.'' Please specify the exact Krippendorff’s Kappa score obtained and the threshold used for ``poor agreement''. Additionally, how do you justify that consistent summary statistics validate the use of these annotator models?

- I am concerned whether reducing dark pattern defined by the authors could unintentionally drop the performance in some popular use cases of LLMs. For instance, social scientists use LLMs to simulate human samples for scientific purposes, using LLMs to generate human-like samples when generating hypotheses or collecting human data is expensive (see Argyle, L. P., Busby, E. C., Fulda, N., Gubler, J. R., Rytting, C., & Wingate, D. (2023). Out of one, many: Using language models to simulate human samples. Political Analysis, 31(3), 337-351. or Törnberg, P., Valeeva, D., Uitermark, J., & Bail, C. (2023). Simulating social media using large language models to evaluate alternative news feed algorithms. arXiv preprint arXiv:2310.05984.). I'm concerned if reducing Anthropomorphization could affect the abilities of LLMs to accurately simulate human agents for scientific purposes.

---

> ### Author Response · Authors · 2024-11-24
>
> Thank you for your comments!
>
> > The authors use LLMs to annotate dark patterns. However, LLMs’ own dark patterns may affect their ability to annotate dark patterns. For instance, if an LLM displays brand bias, it may evaluate responses from its own brand more favorably. A simple statistical test for potential biases in annotation could address this (e.g., comparing whether an LLM's scores for its own responses differ significantly from those it assigns to other LLM responses).
>
> We acknowledge this as a valid concern and have attempted to mitigate the bias by employing three annotator models instead of one. We conducted a statistical test to compare a given annotator's mean score for its own model family versus other models, relative to differences observed among other annotators which is added in the rebuttal revision. We measured the differences between annotators’ ratings, as an annotator assigning its own model family significantly lower scores could be valid if the other annotators also agree. Our findings show that annotators are generally consistent in rating how a given model family compares to others. However, we recognize potential cases of annotator bias, such as in the *brand_bias* category, where the Gemini annotator rates its own model family’s outputs as less deceptive compared to GPT and Claude annotators. We have included additional analyses in Figure 6 of the Appendix.
>
> > The paper lacks detailed information on human annotations, particularly regarding the annotators' demographics or level of expertise. For instance, it would be helpful to clarify whether LimeSurvey annotators were laypeople or experts and whether they reflect a diverse demographic range (age, gender, etc.) similar to typical LLM users.
>
> Thank you for pointing this out. The human annotations were conducted by the authors, who have professional experience with dark patterns. While we acknowledge the possibility of human errors in the annotation process, a detailed rubric for annotators and examples of human-rated dark patterns are included in the Appendix.
>
> > There is no evidence of stability for the benchmark findings across variations in prompt designs. You could test for consistency by paraphrasing prompts in Table 1 and replicate the experiments.
>
> We recognize this as a critical concern, and our benchmark construction process included careful reviews and rephrasings to ensure prompt quality and variability.
>
> > I do not understand 13p line 685: "Despite a low Krippendorff’s Kappa, indicating poor inter-rater agreement, the summary statistics over models and dark patterns remain consistent." Please specify the exact Krippendorff’s Kappa score obtained and the threshold used for poor agreement. Additionally, how do you justify that consistent summary statistics validate the use of these annotator models?
>
> Thank you for highlighting this issue. We have added a table in the Appendix for specific agreement metrics between annotator models and human raters. Our analysis revealed low Kappa scores for certain models and categories (e.g., Gemini annotator and sycophancy). However, the comparative rankings of dark scores remain consistent across annotators, such as Claude 3 family consistently showing the lowest scores, while Llama 3 70B and GPT-3.5-Turbo exhibit the highest scores.
>
> > I am concerned whether reducing dark patterns defined by the authors could unintentionally drop the performance in some popular use cases of LLMs. For instance, social scientists use LLMs to simulate human samples for scientific purposes, using LLMs to generate human-like samples when generating hypotheses or collecting human data is expensive (see Argyle, L. P., Busby, E. C., Fulda, N., Gubler, J. R., Rytting, C., & Wingate, D. (2023). Out of one, many: Using language models to simulate human samples. Political Analysis, 31(3), 337-351. or Törnberg, P., Valeeva, D., Uitermark, J., & Bail, C. (2023). Simulating social media using large language models to evaluate alternative news feed algorithms. arXiv preprint arXiv:2310.05984.). I'm concerned if reducing anthropomorphization could affect the abilities of LLMs to accurately simulate human agents for scientific purposes.
>
> Thank you for referencing these papers. We agree that reducing anthropomorphization may negatively impact the ability of LLMs to simulate human agents for scientific purposes. However, we believe that public, user-facing LLMs should not be designed to be highly anthropomorphic by default. Such capabilities should only be activated if explicitly prompted or fine-tuned to simulate human personas, as described in the referenced works.

---

> ### Comment · Reviewer_xbiZ · 2024-11-26
>
> I'd like to thank the authors for their responses. However, the low inter-rater reliability (as low as 0.4–0.5) for some categories and models remains a concern. This suggests significant disagreement between human and LLM annotators on whether a particular response constitutes a dark pattern.
>
> While the LLM-level average score (i.e., summary statistics) appears consistent (e.g., Claude consistently scoring low), I think this does not mean that low reliability is acceptable. For example, consider the following scenario where a human annotator and an LLM annotator rate three responses from a particular LLM:
>
> | LLM Response | Human Annotator | LLM Annotator |
> |----------|-----------------|---------------|
> | A        | Yes             | No            |
> | B        | No              | Yes           |
> | C        | Yes             | Yes           |
>
> In this case, although the summary statistics (e.g., the "Yes" rate of 66%) are consistent between the human and the LLM annotator, the inter-rater agreement is very low (33%). Such misalignment may suggest that the definition of the concept is not sufficiently clear or unambiguous for the annotators to generate consistent labels, which could be a problem (Or, human and LLM annotator may have different criteria of deciding whether a particular response is a dark pattern). As this work lays the foundation for proposing a new benchmark, I believe that ensuring reliability and validity is very critical to its impact.

---

> > ### Author Response · Authors · 2024-12-01
> >
> > Thank you for your critique regarding inter-rater reliability and summary statistics. To address this, we added the Jaccard index (Table 3) to measure overlap in positive annotations. In scenarios like the one you provided, the Jaccard index would capture misalignment with a low score (e.g., 0.33). Our analysis shows indices exceeding 0.60 for most categories, indicating substantial agreement, while **Brand Bias** (J = 0.40) highlights challenges in subjective categories.
> >
> > Combined with Cohen’s Kappa, Jaccard strengthens our evaluation of annotation consistency and demonstrates the benchmark’s robustness across most categories. We believe this addresses your concerns effectively.

---

> > > ### Comment · Reviewer_xbiZ · 2024-12-01
> > >
> > > I thank the authors for clarifying this. I also notice that the inter-rater reliability is very high for certain categories (e.g., harmful generation; K = .98). I hope the authors elaborate on this further in the paper—e.g., by discussing which categories had the highest reliability, which had lower reliability, possible reasons for these differences, and potential directions for improvement in future work. I have increased my score accordingly.

---

### Public Comment · ~harshraj_bhoite1 · 2025-02-26
**Benchmarking Dark Patterns in LLMs: A Review of DarkBench**

Summary:
This paper presents DarkBench, a benchmark developed to detect and assess dark design patterns in large language models (LLMs). The authors outline six distinct manipulative behaviors and evaluate 14 different LLMs, including both proprietary and open-source models, to analyze their susceptibility to these patterns.

Soundness: 4 (Well-founded)Presentation: 3 (Clear and structured)Contribution: 3 (Relevant and useful)

Strengths:

The paper addresses a significant yet often overlooked concern in AI ethics by thoroughly investigating dark patterns in LLM interactions.

The inclusion of 14 models allows for a broad comparison across various AI providers, enhancing the study’s relevance.

The categorization of dark patterns is systematically designed and supported by existing research, adding depth to the discussion.

Weaknesses:

The use of LLMs for annotation poses a potential risk of bias, as models may evaluate their own outputs more favorably. Incorporating human validation or statistical bias checks would improve credibility.

Additional information on the human annotation process, such as the annotators' expertise and demographic diversity, would enhance transparency.

The study does not explore the stability of benchmark results across different variations of prompts. Testing with rephrased prompts would help ensure the robustness of findings.

The results section would benefit from more concrete qualitative examples to illustrate dark patterns identified in model responses.

Questions:

The paper states in line 685: "Despite a low Krippendorff’s Kappa, indicating poor inter-rater agreement, the summary statistics over models and dark patterns remain consistent." What specific Krippendorff’s Kappa score was obtained, and what threshold defines poor agreement? How does the consistency in summary statistics validate the use of these annotation models?

Could reducing dark patterns inadvertently impact the effectiveness of LLMs in specific applications? Social scientists, for example, use LLMs to simulate human behavior for research purposes (see Argyle et al., 2023; Törnberg et al., 2023). Would minimizing Anthropomorphization negatively affect their ability to model human-like interactions?

Flag For Ethics Review: No ethics review required.Rating: 7 (Valuable contribution, merits consideration)Confidence: 4 (Well-informed assessment, though certain aspects may need further clarification)Code Of Conduct: Yes

---

### Meta-Review · Area_Chair_xvXA · 2024-12-18

**Metareview:**

This paper presents a new benchmark— DarkBench — for evaluating dark patterns in LLMs. The authors develop the benchmark using few-shot prompting, resulting in a total of 660 adversarial prompts across six categories: brand bias, user retention, sycophancy, anthropomorphism, harmful generation, and sneaking. Using a mixture of human annotation and model annotations, the authors evaluate 14 open-sourced and preparatory models on the DarkBench and find prevalence of dark patterns across all models.

This is a timely and novel contribution towards tackling dark patterns in LLMs and to the evaluation of model safety in general. Given how arduous and complex building such a benchmark is, this is a thoughtful and significant contribution. The writing is concise and the figures are clear.

**Additional Comments On Reviewer Discussion:**

A number of issues were raised by reviewers:

Methodology: this includes vague methodology, lack of detailed information on human annotations, and low inter-rater reliability for some categories between the human and the LLM annotator.

The authors address these by providing detailed textual information and added tables for further clarity.


Missing literature: missing relevant work on dark patterns was acknowledged and incorporated.


Theoretical grounding: deviation of “dark patterns” from traditional definitions. The authors provide a satisfactory justification for their specific approach on dark patterns that somewhat deviates from while building on traditional dark patterns work.

Overall, a great contribution!

---

### Decision · Program_Chairs · 2025-01-22

Accept (Oral)